# Insulin Receptor-Related Receptor Regulates the Rate of Early Development in *Xenopus laevis*

**DOI:** 10.3390/ijms23169250

**Published:** 2022-08-17

**Authors:** Daria D. Korotkova, Elena A. Gantsova, Alexander S. Goryashchenko, Fedor M. Eroshkin, Oxana V. Serova, Alexey S. Sokolov, Fedor Sharko, Svetlana V. Zhenilo, Natalia Y. Martynova, Alexander G. Petrenko, Andrey G. Zaraisky, Igor E. Deyev

**Affiliations:** 1Shemyakin-Ovchinnikov Institute of Bioorganic Chemistry, Russian Academy of Sciences, 117997 Moscow, Russia; 2European Molecular Biology Laboratory, European Bioinformatics Institute, Hinxton CB10 1SD, UK; 3Federal Research Center “Fundamentals of Biotechnology”, Russian Academy of Sciences, 119991 Moscow, Russia; 4Scientific Center for Genetics and Life Sciences, Sirius University of Science and Technology, Olimpiyskiy prosp. 1, 354340 Sochi, Russia

**Keywords:** insulin, alkaline pH, embryogenesis, tyrosine kinase receptor, transcriptome

## Abstract

The orphan insulin receptor-related receptor (IRR) encoded by *insrr* gene is the third member of the insulin receptor family, also including the insulin receptor (IR) and the insulin-like growth factor receptor (IGF-1R). IRR is the extracellular alkaline medium sensor. In mice, *insrr* is expressed only in small populations of cells in specific tissues, which contain extracorporeal liquids of extreme pH. In particular, IRR regulates the metabolic bicarbonate excess in the kidney. In contrast, the role of IRR during *Xenopus laevis* embryogenesis is unknown, although *insrr* is highly expressed in frog embryos. Here, we examined the *insrr* function during the *Xenopus laevis* early development by the morpholino-induced knockdown. We demonstrated that *insrr* downregulation leads to development retardation, which can be restored by the incubation of embryos in an alkaline medium. Using bulk RNA-seq of embryos at the middle neurula stage, we showed that *insrr* downregulation elicited a general shift of expression towards genes specifically expressed before and at the onset of gastrulation. At the same time, alkali treatment partially restored the expression of the neurula-specific genes. Thus, our results demonstrate the critical role of *insrr* in the regulation of the early development rate in *Xenopus laevis*.

## 1. Introduction

Receptor tyrosine kinases are among the most important members of the cell signaling system. They play a key role in the organism development and functioning, and participate in the regulation of the cell interactions, cell proliferation and differentiation, cell migration and metabolism, and control of the cell cycle. To activate most of the receptor tyrosine kinases, a ligand of protein or peptide nature is necessary, which is capable of binding to the receptor simultaneously in several sites, causing its dimerization and activation, resulting in autophosphorylation of the intracellular part of the receptor. An exception is the insulin receptor family, whose members are dimerized by disulfide bonds in an inactive state, and interaction of their extracellular part with the ligand leads to conformational changes followed by their activation [1]. This family includes three receptors: insulin receptor (IR), insulin-like growth factor receptor (IGF1R) and insulin receptor-related receptor (IRR). For the IRR, unlike its homologs, ligands of peptide-protein nature were not found [2]. All three receptors are highly homologous and, presumably, have similar mechanisms of functioning.

Unexpectedly, we found that human IRR can be activated by alkaline extracellular medium [3,4]. Activation of IRR by alkali is specific, reversible, and dose-dependent [4]. IRR orthologs from frog and mouse also can be activated by alkaline medium. Activation of endogenous IRR by alkali in insulinoma MIN6 cells leads to IRS-1 phosphorylation and cytoskeleton remodeling in these cells [5].

Expression of the *insrr* in mice was detected in the stomach, kidneys, and pancreas in adult animals. All these organs are involved in excretion or secretion of fluids with an alkaline pH [2]. Mice with *insrr* knockout did not show gross phenotype under normal conditions [6]. However, the in vivo analysis of *insrr* knockout mice phenotype under experimentally induced alkalosis revealed the role of IRR as a regulator of bicarbonate excretion in the kidneys [4,7]. Also, *insrr* is expressed in the alpha and beta cells of the islets of Langerhans, in enterochromaffin-like cells of the stomach, and in certain types of neurons [6,8,9,10,11,12,13,14]. The expression of *insrr*, as well as other members of the IR family (*insr* and *igf1r*), was found at the stage of unicellular mouse embryos and blastocysts [15]. Lately, IRR is found to be a potential therapeutic target for heart cardioprotection [16]. Our recent studies have revealed a possible role for the receptor tyrosine kinase IRR in the development of mouse preimplantation embryos. Using the Mouse Embryo Assay (MEA) we showed that blastocyst yield for *insrr* knockout animals was significantly lower than for wild-type mice: 6.7% and 43.8% of the total number of isolated zygotes, respectively [17].

Based on phylogenetic analysis of the IR family sequences, *insrr* is present in amphibians remaining conserved with the mammalian orthologs [18]. According to the transcriptome analysis, *insrr* ortholog in *Xenopus laevis* and *Xenopus tropicalis* frogs is highly expressed in embryogenesis before neurulation [19,20]. One gene from the IR family was found in flies and worms, and only two orthologs of IR and IGF1R were found in fish [18,21]. In the present work, we studied the physiological role of *insrr* in early development of *Xenopus laevis* using morpholino-mediated knockdown. As a result, we showed that downregulation of *insrr* leads to delay in development, accompanied by a shift in gene expression towards earlier expressing genes. At the same time, incubation of the embryos in the alkaline medium eliminated these effects.

## 2. Results

### 2.1. Insrr Knockdown Elicits Retardation of Embryonic Development

To understand the role of *insrr* in *Xenopus laevis* development, we downregulated its expression by injecting morpholino antisense oligonucleotide (*insrr* MO1) into early embryos (Figure 1). As a result, a significant retardation of development was observed in these embryos when comparing to their siblings injected with the *control* MO (Figure 1A,B). The first signs of such retardation became visible only during neurulation, when retardation of the neural tube closure was seen in embryos injected with *insrr* MO1 compared to control siblings. Interestingly, most of the embryos with downregulated *insrr* and demonstrated developmental delay between the late neurula stage and tadpole stages 28–32, then gradually became equal in appearance with the control siblings (Figure 1A).

Since IRR is known as a receptor for which activity is modulated by alkaline pH, we decided to test effects of alkaline medium on the development of embryos with downregulated *insrr*. To this end, we incubated embryos injected with *insrr* MO1 and with the *control* MO in solutions with pH 7.2 and 8.5. As a result, we revealed that whereas the development of the *insrr* MO1-injected embryos was significantly delayed compared to the control siblings at pH 7.2, no such difference was detected for *insrr* MO1-injected embryos incubated at pH 8.5 (Figure 1C,D). In other words, alkaline pH medium “rescued” embryos with downregulated *insrr* from the retardation in development. It is noteworthy that in wild-type embryos grown in an alkaline medium (pH 8.5), no delay or acceleration of development was observed compared to control (pH 7.2).

Importantly, the specificity of the development retardation effect elicited by the *insrr* downregulation was confirmed by the fact that *insrr* MO2, designed to another target site on *insrr* mRNA, elicited a similar effect to *insrr* MO1 (Appendix A). Besides, we confirmed specificity of MO1 by demonstrating that the retardation effect caused by this MO could be rescued if *insrr* mRNA deprived of the MO1 target site was co-injected (Appendix A). In addition, we proved efficiency of the MO suppressive effect upon translation of both *insrr*.*L* and *insrr*.*S* homeolog mRNAs, detecting N-fragments of proteins encoding by these mRNAs by Western blotting (see Material and Methods and Appendix A).

### 2.2. Insrr Knockdown Causes a Shift of Gene Expression towards Genes Expressed in Earlier Stages of Development

As shown by PCA analysis, bulk RNA-seq data obtained for transcriptomes of all four groups of embryos clearly differed from each other (Figure 2A). In comparison with control group at pH 7.2, 1650 differentially expressed genes (DEG) were detected in transcriptome after *insrr* MO1 (designated as MO below) injection at pH 7.2 (padj < 0.05), including 807 upregulated and 843 downregulated genes. In case of alkali treatment of embryos injected with *insrr* MO, we found 1235 DEGs (padj < 0.05), including 635 upregulated and 600 downregulated genes (Figure 2B). Also, comparison of transcriptomes of the control group at pH 8.5 and embryos injected with *insrr* MO1 at pH 8.5 revealed 1519 DEGs (padj < 0.05), including 770 upregulated and 749 downregulated genes (Figure 2B, Appendix A). Interestingly, injection of this morpholino led to some increase of the *insrr* mRNA level, with log2(FoldChange) 0.647 at pH 7.2 and with log2(FoldChange) 0.513 at pH 8.5 (Appendix A). This effect is not surprising and was observed earlier after injections of morpholinos to mRNA of other genes. In particular, we detected a similar effect in the case of blocking by morpholino the translation of mRNA of the homeobox gene *Xanf* [22]. This effect likely indicates a negative feedback regulatory loop between the protein concentration and the level of transcription of its own gene.

To visualize the action of alkali we made graphic analysis of transcriptomes of the control and *insrr* MO-injected embryos at pH 7.2 depending on the degree of change in the expression of the genes. The density plot on a field of log2(FoldChange) vs. log10(baseMean) was plotted and genes that pass a threshold of padj < 0.05 in differential expression analysis between wild-type and *insrr* knockdown are colored red (Figure 2C). Other genes are colored grey. Then, using multicolor indication, we labeled DEGs in the same plot, which were downregulated (Figure 2D) and upregulated (Figure 2E) in *insrr* MO-injected embryos under alkali treatment. We found that under alkaline treatment of *insrr* knockdown embryos, most downregulated genes had increase in their expression, and vice versa, upregulated genes had decreased expression.

Then, we annotated a list of DEGs in the *insrr* MO-injected embryos and control MO siblings at pH 7.2 (*insrr* MO 7.2/control MO 7.2 DEGs) (Appendix A). Upregulated DEGs were mostly enriched in microtubules and mitotic-cycle-regulation genes groups. Downregulated DEGs were found in RNA/DNA binding, ribosome, and metabolic regulation genes groups. Similar analysis was made for DEGs in *insrr* MO-injected embryos grown at pH 7.2 and pH 8.5 (*insrr* MO 8.5/*insrr* MO 7.2) (Appendix A). We found annotations only for upregulated DEGs in RNA binding, ribosome and gene expression groups (Appendix A). Among top genes with changed expression after *insrr* MO injection are actin and myosin coding genes, transcription factor pax6 and cycline ccna2 (Appendix A).

Interestingly, by analyzing temporal expression patterns of the15 most upregulated and most downregulated *insrr* MO 7.2/control MO 7.2 DEGs (Table 1) using the Xenbase online database, we revealed the following regularity consistent with the morphological retardation of development. Namely, in the wild-type embryos, these upregulated genes had maximum levels of their expression very early, either before or at the very beginning of gastrulation, and the expression of the downregulated genes reached its maximum levels much later: only after the beginning of neurulation (see a representative example of the temporal expression profiles of two DEGs from these groups in Figure 3A and a full table of profiles for all 30 selected DEGs in Appendix A). At the same time, when we had checked these 30 genes among *insrr* MO 8.5/*insrr* MO 7.2 DEGs, we found that under the influence of an alkaline medium, all of them had had a change in their expression levels to the opposite (Table 1).

For a more comprehensive analysis of the effect elicited by *insrr* knockdown and its alkali rescuing, we compared transcriptomes of *Xenopus laevis* embryos at stage 15 with downregulated *insrr* and their wild-type siblings grown under neutral (pH 7.2) and alkaline (pH 8.5) conditions as described in Materials and Methods.

To test if the revealed regularity could be also true in relation to other *insrr* MO 7.2/control MO 7.2 DEGs, we built the point diagram, plotted log2(Fold change *insrr* MO 7.2/control MO 7.2) versus log2(FoldChange stage10/stage15) (Appendix A) for each of 1461 DEGs (Figure 3A), which were found in intersection of 1650 DEGs after *insrr* MO injection at pH 7.2 and 42,638 genes expressed during gastrulation [19] (Figure 3B,C). This diagram shows that most DEGs have log2(Fold change MO7.2/WT7.2) values comparable and similar as have log2(FoldChange stage10/stage15) values for the same genes. In other words, for most DEGs, the rule is true that if DEG has a higher expression level in embryos with downregulated *insrr* compared to the wild-type siblings, then one may expect that this gene should demonstrate a higher expression level at stage 10 compared to stage 15 during normal development. And vice versa, if DEG has a lower expression level in embryos with downregulated *insrr*, this indicates that this gene should demonstrate a lower expression level at stage 10 compared to stage 15. In sum, this confirms the overall validity of the regularity, suggesting a shift in gene expression towards earlier expressed genes in embryos with downregulated *insrr*.

### 2.3. The Effect of Alkaline Medium on Gene Expression in Wild-Type and Insrr Knockdown Embryos Is Different

To test if alkaline treatment indeed could massively invert changes of expression revealed for *insrr* MO 7.2/control MO 7.2 DEGs, we selected 331 DEGs that were at the intersection of 929 *insrr* MO 7.2/control MO 7.2 DEGs (padj < 0.01) and 634 *insrr* MO 8.5/*insrr* MO 7.2 DEGs (padj < 0.01) (Appendix A) and compared their expression levels in conditions of *insrr* MO 7.2/control MO 7.2 and *insrr* MO 8.5/*insrr* MO 7.2 experiments. As one may see in the point diagram in Appendix A, direction of changes of expression levels of all these 331 DEGs after incubation of embryos in alkaline medium (pH 8.5) was truly the opposite to that observed in the neutral medium (pH 7.2). Importantly, the majority of these 331 DEGs sensitive to alkaline pH in *insrr* knockdown embryos are the genes that have the greatest change in expression in *insrr* knockdown embryos grown in the neutral medium (Table 1 and Appendix A). Taking into account the known modulating effect of alkaline pH medium on IRR activity, one may conclude that this correlation once again confirms the specificity of changes in gene expression observed in embryos with downregulated *insrr*. Also, we annotated the list of the selected 331 DEGs and found them in ribosome, translation, and metabolic regulation genes groups (Appendix A).

In addition, we investigated differentially expressed genes in the wild-type *Xenopus laevis* embryos grown at pH 8.5 and pH 7.2 (control MO 8.5/control MO 7.2 DEGs). Alkali treatment of embryos resulted in 849 DEGs (padj < 0,05). Of those, 499 genes were upregulated, and 350 genes downregulated (Figure 2B). Upregulated genes include factors involved in adherent junction, WNT signaling pathway (*lef1*, *smad4*), insulin signaling (*insr*, *irs1*), cell cycle (*cdc27*, *cdk4*, *bub1*), RNA metabolic process, regulation of gene expression and other processes (Appendix A). Genes with a reduced level of expression after alkali treatment were associated with spliceosome, embryo development, BMP signaling pathway and other processes (Appendix A).

Interestingly, the group of common genes between WT8.5/WT7.2 and MO8.5/MO7.2 DEGs almost does not intersect with the group of common genes between MO7.2/WT7.2 and MO8.5/MO7.2 DEGs. Thus, in wild-type embryos, the alkaline medium regulates genes other than those whose expression it restores after *insrr* knockdown.

## 3. Discussion

The physiological role of IRR in pH maintenance in the renal system and in regulation of some behavior traits was previously demonstrated in adult mice [2,23]. Although *insrr* knockout mice did not show defects in early embryonic development in vivo, blastocyst yield in knockout animals is significantly lower than in wild-type animals according to the results of in vitro analysis, suggesting a possible role for the receptor tyrosine kinase IRR in the development of mouse preimplantation embryos [17]. Acid-base balance regulation in frogs is different from mice or humans. For example, blood pH above 8.0 is usually observed depending on habitat and temperature [24,25], and due to this the role of IRR as an alkali sensor can be more important in frogs than in mice or humans. Also, frogs and toads can live and breed in a wide range of water pH, from acidic (pH about 4.0) to alkaline (pH about 10.0) [26], which indicates the presence of a mechanism for regulating the expression of various genes depending on the external pH.

We now studied the effects of temporary downregulation of *insrr* functioning in *Xenopus laevis* embryos. As a result, the following set of data, schematically shown in Figure 4, was obtained, indicating that IRR may regulate the rate of early development in frogs. Downregulation of *insrr* by anti-sense morpholinos resulted in a significant morphological retardation of embryo development. The specificity of this effect is confirmed by its reversibility after alkaline treatment of the *insrr* knockdown embryos, as well as by the rescue experiments, in which *insrr* MO was co-injected with *insrr* synthetic mRNA which lacked the MO target site.

In addition to the morphological retardation of development observed at the tadpole stages, we revealed in early neurula embryos with downregulated *insrr* a massive expression shift from later expressed genes towards earlier ones. Thus, in the first group were genes whose expression is increased during neurulation and which are responsible mainly for skeletal muscle and neural tube differentiation, including *desmin*, several genes involved in adrenergic signaling, *foxg1*, several genes of *hox* compex, *mylfp*, *myl1*, *pax6*, *pitx2* and others. Among the second group were genes regulating overall patterning of the embryo before and at the very beginning of gastrulation: *cer1*, *eomes*, *frzb*, *hhex*, *mixer*, *pou5f3*, and many others. Importantly, as in the case of morphological retardation of development, expression of these genes could be temporarily rescued by incubation in alkaline pH medium.

Previous results revealed no developmental abnormalities of *insrr* genetic knockout mice. However, the developmental retardation observed in Xenopus embryos with downregulated *insrr* simply could be overlooked in mice due to the difficulty of observing intrauterine development, especially at early stages. In addition, there is growing evidence that in many cases genetic knockout in mice may not be phenotypic, due to compensatory mechanisms that bypass the knockout effects by activating homologous genes that can substitute the function of the knockout gene [27,28]. In particular, there is evidence that the overall rate of development also can be regulated by signaling through an insulin receptor [29,30]. Finally, the mechanisms of early development in mice and frogs could be different in aspects concerning IRR signaling.

The developmental retardation revealed in embryos with downregulated *insrr* was accompanied by a synchronous temporal shift in the expression of many genes that coordinately expressed in normal embryogenesis but are generally not mechanistically connected with each other. This indicates that signaling through IRR may regulate some chromatin remodeling, governing simultaneous switching between large gene ensembles. It can be assumed that such remodeling could most likely be associated with chromatin acetylation, which is known as a critical epigenetic factor responsible for altering the accessibility of promoters of genes at successive developmental stages [31,32,33]. Thus, histone hyperacetylation is essential to enhance accessibility for the transcription machinery of the promoters of genes normally expressed before the early gastrula stage, while promoters of genes expressed during neurulation and responsible for the muscle and neural crest differentiation do not need hyperacetylation [34].

The level of chromatin acetylation is determined by the balance between histone acetylases (HATs) and deacetylases (HDACs) activities [35]. As downregulation of *insrr* shifts the embryonic expression towards genes with maximum expression in the early stages, which are characterized by chromatin hyperacetylation, one may suppose that IRR signaling either inhibits HATs or stimulates HDACs activities. Indeed, in both of these cases, suppression of IRR signaling would lead to chromatin hyperacetylation and a shift in expression towards earlier expressed genes. However, the concentration of *insrr* transcripts is maximal at the beginning of embryogenesis and then decreases to a very low level by the onset of neurulation [19]. Such a temporal profile of *insrr* expression contradicts the assumption of its direct suppressive effect on HAT activity at developmental stages before gastrulation since these stages are characterized precisely by hyperacetylation of chromatin. Thus, it seems more likely that IRR signaling may not inhibit HATs but stimulate HDACs activity. In agreement with this, it was shown that treatment of embryos by butyrate, a noncompetitive inhibitor of HDACs [36], elicits retardation of development phenotypically similar to that we observed in the case of *insrr* downregulation by MO [32].

In a continuation of this, one may suppose that since IRR is the receptor tyrosine kinase, it could stimulate the activity of HDACs by regulating their phosphorylation. Indeed, it was shown, for example, that phosphorylation of HDAC1 and HDAC2 by protein kinase CK2 increases their activity by several times [37,38]. Furthermore, the activity of protein kinase CK2 itself is stimulated by phosphorylation of its two tyrosine residues [39]. In addition, other protein kinases capable of HDACs phosphorylation, for example, cAMP-dependent protein kinase and protein kinase G in the case of HDAC1 [37], can be also considered as potential targets for IRR signaling in early embryonic cells. Obviously, all these speculations could be tempting topics for further experimental testing.

Interestingly, whereas we revealed clear retardation of development and rescuing effect of alkaline pH or synthetic *insrr* mRNA injected in embryos with downregulated *insrr*, we did not see visible changes in the developmental rate of the wild-type embryos treated by alkaline or acidic pH. Also, no acceleration of development compared to the wild-type embryos was detected in their siblings injected with synthetic *insrr* mRNA (data not shown). These results indicate that in the wild-type embryos neither pH nor the concentration of IRR is the limiting factor that determines overall signaling intensity through this receptor. Otherwise, it would be difficult to explain why only a strong decrease in the concentration of IRR in embryos injected with MO, but not pH or overexpression of *insrr*, allowed us to reveal the role of this receptor in regulating the rate of embryonic development (Figure 4). In turn, this may indicate that there are some other proteins in embryo, which interact with IRR and play a role of limiting factor in the regulation of IRR signaling. Furthermore, it can be assumed that although the protein ligand of IRR has not been found, this ligand may exist, and it regulates IRR signaling in Xenopus embryos. In this regard, early Xenopus embryos could be considered as a promising model for searching for IRR interacting proteins. Also, it should be noted that other membrane proteins such as alkaline pH-sensing channels (for example, TASK-2) [40] or alkaline pH activated receptors (Erbb2, c-Met) [41,42] may be involved in the action of extracellular alkaline pH medium on *insrr* knockdown in Xenopus laevis embryos.

## 4. Material and Methods

### 4.1. Embryo Manipulations

All experiments with animals were performed according to the protocol of the Institutional Animal Care and Use Committee (IACUC) approved by the Bioethics Commission of the Shemyakin-Ovchinnikov Institute of Bioorganic Chemistry of the Russian Academy of Sciences (IBCH RAS) and handled in accordance with the Animals (Scientific Procedures) Act 1986 and Helsinki Declaration. For microinjection experiments, the *Xenopus laevis* embryos were obtained, fertilized in vitro and dejellied in 2% cysteine in 0.1 × MMR (Marc’s Modified Ringer) solution. The microinjections were performed in 4% Ficoll in 0.1 × MMR. The incubation of embryos at alkaline pH was performed in 0.1 × MMR solution adjusted by NaOH to pH 8.4. Upon reaching stage 26–27, embryos were photographed on Leica M205 stereomicroscope.

### 4.2. Morpholino and mRNA Injection Experiments

The morpholino oligonucleotides for *insrr* mRNA were ordered in Gene Tools, LLC: *Insrr* MO1: 5’-CATGATATGAAGACGCCAACATTAA-3′ (complementary to nucleotides −22 to 3 of the *insrr* mRNA); *insrr* MO2: 5’-AAAACTTGTCTGTCCATGATATGAA-3′ (complementary to nucleotides −8 to +17 of the *insrr* mRNA). The control MO: 5′-CAAAATACCAAGAAACCAAGGTTAA-3′. For microinjections, MOs were diluted to the final concentration of 0.3 mM and mixed with fluorescein-labelled lysinated dextran for tracing. Embryos at 2-cell stage were injected with 6 nl of MO solution into each blastomere.

To test MO specificity, we performed the phenotype rescue experiments by co-injection of MO with *insrr* mRNA [3]. For mRNA synthesis, pCS2-*insrr* plasmid encoding the full-length xenopus IRR was linearized with *Not*I restriction endonuclease, and the capped mRNA was synthesized using the SP6 mMESSAGE mMACHINE Transcription kit (Ambion). Then, we added synthetic *insrr* mRNA at final concentration 5 ng/µL into MO1 mixture for injection. Embryos injected with this mixture were compared with embryos injected with MO1 and control MO.

For RNA-seq, embryos injected with MO1 or with the control were incubated at pH 7.2 and 8.5 till embryonic stage 13 (early neurula) and five embryos of each type were collected in three replicates in RNAlater solution.

For testing MO efficiency, we used myc-tagged 5′ fragments of *insrr* cDNA. To prepare plasmids coding C-myc-tagged N-fragments of *insrr*-L and *insrr*-S, the corresponding cDNA fragments were obtained by PCR with the following pairs of primers (coordinates of primers’ target sites on mRNA sequences are indicated in brackets):

*insrr*. *L* myc cloning dir

5′-CCTACAGATCTGAGGATTTTTC (−46 to −24)

*insrr*. *L* myc cloning rev

5′-aattctcgagcacgcaggcacggggatcat (+727 to +747)

*insrr*. *S* myc cloning dir

5′-TTTACAGATCTGAGGATTTTAT (−54 to −32)

*insrr*. *S* myc cloning rev

5′-aattctcgagcacacaggcacgaggatctc (+724 to +744)

The obtained cDNA fragments, encoding 249 and 248 a.a. length fragments of *insrr*. *L* and *insrr*. *S*, respectively, were sub-cloned using *Eco*RI (blunted) and *Xho*I sites into pCS2-twsg1-myc plasmid (kindly provided by E. Parshina) instead of twsg1 coding region and checked by sequencing. To obtain capped mRNA, these plasmids were linearized by *Acc*65I, and mRNAs were synthesized using mMESSAGEmMACHINE kit (Ambion).

For injection experiments, MOs were diluted to the final concentration of 0.3 mM; mRNAs were diluted to the final concentration of 25 ng/µL. The mRNA or MO solutions were mixed with FLD (Fluorescein Lysinated Dextran, 40 kDa, 5 mg/mL, Invitrogen) and 4–5 nl of the mixture were injected into single blastomeres at two-cell stage. Injected embryos were grown until stage 13, at which myc-tagged IRR fragments in crude lysates of these embryos were analyzed by Western blotting as described in [43]. Coomassie stained gels were used as loading controls.

### 4.3. RNA Purification and RNA-seq

All RNA purifications and RNA-seq analysis were made by Genoanalytica (Moscow, Russia). Briefly, total RNA was extracted from the samples with Trizol reagent according to the manufacturer’s protocol. Quality of RNA was checked with BioAnalyser and RNA 6000 Nano Kit (Agilent). Poly(A) RNA was purified using Dynabeads^®^ mRNA Purification Kit (Ambion). Illumina library was made from poly(A) NEBNext^®^ Ultra™ II RNA Library Prep Kit for Illumina (NEB) according to the manufacturer’s protocol. Sequencing was performed on Illumina HiSeq1500 with 50 bp read length. At least 30 million reads were generated for each sample. Reads were mapped to the genome using Star Aligner and the fold changes of gene expression were calculated using DEseq2.0 software (Appendix A). Raw RNA-seq data were deposited to European Nucleotide Archive under the accession number PRJEB45623.

### 4.4. Gene Expression Analysis

For analysis of our RNA-seq data and previously published RNA-seq data [9], we used identical software. For reads mapping, STAR v2.7 software was used [44]. For reads counting we used HTSeq v0.11.2 software [45], and finally for differential transcriptional and principal component analysis (PCA), DESeq2 v3.9 software was used [46]. Enriched Gene Ontology terms were identified using DAVID bioinformatics resources with genes pre-ranked according to their fold change induced by *insrr* morpholino, pH treatment or both [47,48]. The temporal expression patterns for the selected genes were retrieved from the online resource xenbase.org.

## Figures and Tables

**Figure 1 ijms-23-09250-f001:**
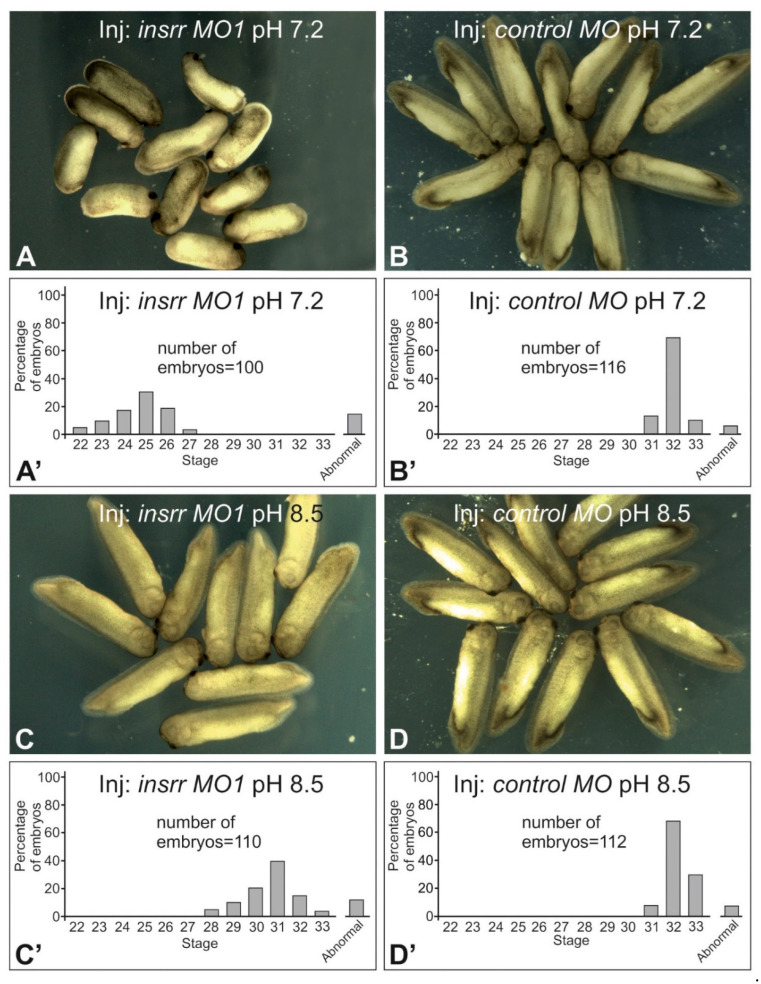
Downregulation of *insrr* by anti-sense morpholino injection retards the *Xenopus laevis* embryos’ development, whereas alkaline pH prevents this effect. (**A**). Embryos after injection of *insrr* MO1 at pH 7.2. (**B**). Embryos after injection of control MO at pH 7.2. (**C**). Embryos after injection of *insrr* MO1 at pH 8.5. (**D**). Embryos after injection of control MO at pH 8.5. The diagrams on (**A’**–**D’**) demonstrate the percentage distribution of embryos, some of which are shown on (**A’**–**D’**), by stage of development. As one may see, knockdown of *insrr* by *insrr* MO1 significantly retards the developmental rate of embryos comparing to their control siblings, and also distribution of the knocked-out embryos developed at pH 8.5 is significantly shifted towards the distribution seen in the wild-type siblings developed at pH 7.2 and pH 8.5.

**Figure 2 ijms-23-09250-f002:**
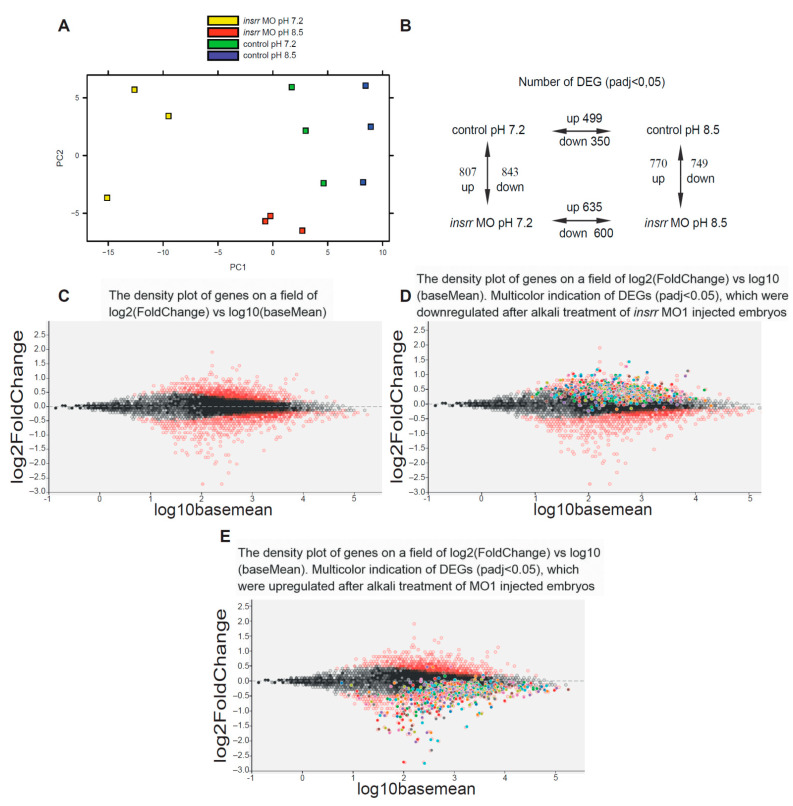
(**A**). PCA analysis of RNA-seq data of control embryos and *insrr* MO1 knockdown embryos after incubation in pH 7.2 or 8.5 media. (**B**). Representation of upregulated and downregulated DEGs (padj < 0.05) of control embryos and *insrr* MO1 knockdown embryos after incubation in pH 7.2 or 8.5 media. (**C**). The density plot of genes on a field of log2(FoldChange) vs. log10(baseMean) was plotted after differential expression analysis of transcriptomes between control and *insrr* MO1 knockdown embryos. DEGs with padj < 0.05 are colored in red and other genes are colored in grey. (**D**). Same plot with multicolor indication of DEGs (padj < 0.05) which were downregulated after alkali treatment of *insrr* MO1-injected embryos. (**E**). Same plot with multicolor indication of DEGs (padj < 0.05), which were upregulated after alkali treatment of *insrr* MO1-injected embryos.

**Figure 3 ijms-23-09250-f003:**
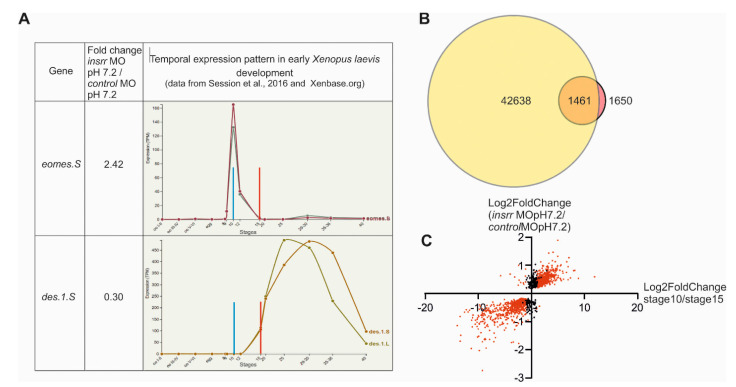
(**A**). Example of the temporal expression profiles of eomes.S and des.1.S genes. Data from [19] and Xenbase.org. (**B**). The intersection of expressed genes from previously published *Xenopus laevis* stage gene expression analysis at 10 and 15 stages and DEGs between transcriptomes of the control and *insrr* MO1-injected embryos at pH 7.2. (**C**). Graphical plot log2 (Fold change *insrr* MO 7.2/control MO 7.2) vs. plotted log2 (FoldChange stage10/stage15) for each 1465 DEGs. Red color indicated only DEGs between stage 10 and stage 15.

**Figure 4 ijms-23-09250-f004:**
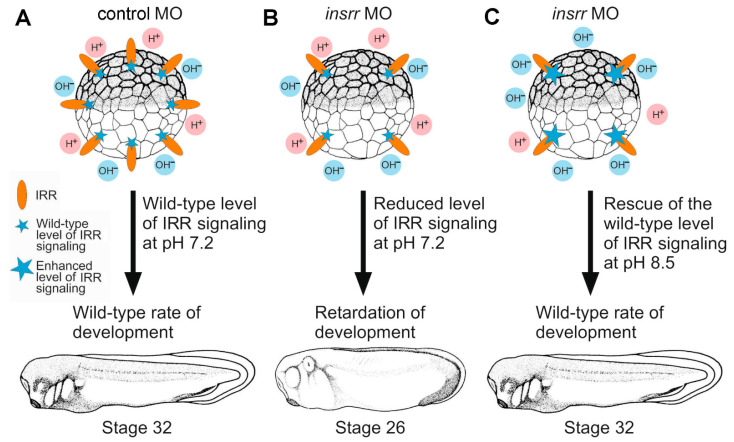
A model of the regulation of embryonic development by IRR. (**A**). In embryos injected with the control MO and at the neutral pH (7.2), the level of IRR signaling is enough to ensure the wild-type rate of embryonic development. (**B**). If *insrr* is downregulated by injection of *insrr* MO, the concentration of IRR molecules significantly but not completely decreases. As a result, at neutral pH (7.2) a decreased level of IRR signaling leads to the retardation of development. (**C**). At alkaline pH (8.5), despite the concentration of IRR molecules being significantly decreased in embryos injected with *insrr* MO, the level of the IRR signaling enhances, which ensures the wild-type rate of embryonic development.

**Table 1 ijms-23-09250-t001:** Top 15 downregulated (red) and 15 upregulated genes (green) after MO1 injection into embryos at pH 7.2.

Gene	Fold Change *insrr* MO 7.2/control MO 7.2	Fold Change *insrr* MO 8.5/*insrr* MO 7.2	
mylpf.L	0.15	2.95	Myosin light chain, phosphorylatable, fast skeletal muscle L homeolog
act3.L	0.15	1.89	Actin alpha 4 L homeolog
act2	0.20	2.32	C-C motif chemokine ligand 4
myl1.S	0.25	1.93	Myosin light chain 1 S homeolog
col2a1.L	0.26	2.07	Collagen, type II, alpha 1 L homeolog
des.1.L	0.26	2.11	Desmin, gene 1 L homeolog
ankrd37.L	0.29	1.12	Ankyrin repeat domain 37 L homeolog
fbxl22.S	0.30	1.94	F-box and leucine-rich repeat protein 22 S homeolog
tnnc2.L	0.30	1.68	Troponin C2, fast skeletal type L homeolog
pax6.S	0.30	2.43	Paired box 6 S homeolog
des.1.S	0.30	1.95	Desmin, gene 1 S homeolog
smyd1.L	0.31	2.21	SET and MYND domain containing 1 L homeolog
tnnt3.L	0.32	2.26	Troponin T type 3, fast skeletal type L homeolog
nr2f5.S	0.32	1.87	Nuclear receptor subfamily 2, group F, member 5 S homeolog
thbs4.S	0.32	1.85	Thrombospondin 4 S homeolog
ccna1.L	1.99	0.48	Cyclin A1 L homeolog
cdc6.L	1.99	1.11	Cell division cycle 6 L homeolog
neu1.L	2.00	0.56	Neuraminidase 1 L homeolog
cdk5r2.S	2.03	0.57	Cyclin-dependent kinase 5, regulatory subunit 2 (p39) S homeolog
mapre3.S	2.05	1.01	Microtubule associated protein RP/EB family member 3 S homeolog
cer1.S	2.11	0.54	Cerberus 1, DAN family BMP antagonist S homeolog
patl2.L	2.18	0.86	Protein associated with topoisomerase II homolog 2 L homeolog
ccna2.L	2.18	0.85	Cyclin A2 L homeolog
xpo6.S	2.19	0.67	Exportin 6 S homeolog
tapt1.S	2.25	0.67	Transmembrane anterior posterior transformation 1 S homeolog
eomes.S	2.42	0.36	Eomesodermin S homeolog
zpd.L	2.51	1.05	Zona pellucid protein D L homeolog
frzb.S	2.71	0.39	Frizzled-related protein S homeolog
fitm2.L	2.77	0.94	Fat storage-inducing transmembrane protein 2 L homeolog
larp6-like.1.S	3.71	0.97	Acheron S homeolog

## Data Availability

Raw data are publicly available.

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
