# Peer review of "Insulin Receptor-Related Receptor Regulates the Rate of Early Development in Xenopus laevis"

_ijms, 2022, doi:10.3390/ijms23169250_

Round 1

Reviewer 1 Report

The insulin receptor-related receptor (IRR), encoded by the insrr gene, is a sensor of the extracellular alkaline medium. It is highly expressed in frog embryos. The authors explore its role by morpholino-mediated selective knockdown of insrr mRNA. They demonstrate that this knockdown leads to delayed embryonic development, but that this phenotype is rescued by incubation of embryos in alkaline medium. They also show that insrr knockdown shifted gene expression towards genes specifically expressed before and at the onset of gastrulation. Alkali treatment partially restored the expression of the neurula-specific genes.

These findings are new and interesting. However, I find it somewhat counter-intuitive that the phenotype of the knockdown of an alkali sensor is rescued by alkali, the very stimulus of the receptor that has been suppressed. It would be nice to have a measurement of the IRR mRNA levels before and after morpholino treatment. Also, while the IRR is an alkali sensor, it is probably not the only cellular component that can be affected by alkaline pH.

Author Response

Thank you for the valuable comments. Upon the reviewer’s request, we added additional text with analysys of insrr mRNA expression after morpholino injection based on our NGS expression data (p.6).

“Interestingly, injection of this morpholino led to some increase of the insrr mRNA level, with log2(FoldChange) 0,647 at pH 7.2 and with log2(FoldChange) 0,513 at pH 8.5 (supplement file 1). This effect is not surprising and was observed earlier after injections of morpholinos to mRNA of other genes. In particular, we detected a similar effect in the case of blocking by morpholino the translation of mRNA of the homeobox gene Xanf (Ermakova et al., 2007). This effect likely indicates a negative feedback regulatory loop between the protein concentration and the level of transcription of its own gene.

Also, we have added text about other pH-sensitive membrane proteins to the discussion section (p. 11).

Also, should be noted, that other membrane proteins like alkaline pH-sensing channels (for example, TASK-2)(Morton et al., 2003) or alkaline pH activated receptors (Erbb2, c-Met)(O. Serova et al., 2019; O. V. Serova et al., 2019) may be involved in the action of extracellular alkaline pH medium on insrr knockdown in Xenopus laevis embryos.

Reviewer 2 Report

In this manuscript the authors examine the role of insulin receptor-related receptor (IRR) in the early stages of Xenopus laevis' development. Using insrr knockdown in embryos from Xenopus laevis, the authors were able to demonstrate delayed development in the media with pH7.2, which was abolished in pH8.5. Overall, these data uncover new and unexpected role of IRR in the development biology. The manuscript is logically organized, uses adequate methods and quite well written. I have just a couple of minor concerns.

1.    An interesting observation reported on Fig.2B, where the alkali media changes genes expression profile in control embryos. It would be very important to elaborate on this observation a bit more to explain it.

2.    Please provide better quality images for the Fig. 2.

Author Response

We agree to this comment and expand the Discussion with text (p. 9);

“Acid-base balance regulation in frog is different from mice or human. For example, blood pH above 8.0 is usually observed depending on habitat and temperature (Boutilier et al., 1987; Howell et al., 1970) due to this reason the role of IRR as alkali sensor can be more important in frog than in mice or human. Also, frogs and toads can live and breed in a wide range of water pH from acidic (pH about 4.0) to alkaline (pH about 10.0) (Wijethunga et al., 2015), which indicates the presence of a mechanism for regulating the expression of various genes depending on the external pH.”

  1. Please provide better quality images for the Fig. 2.

We added Fig.2 with better quality
